# Validity of Peripheral Oxygen Saturation Measurements with the Garmin Fēnix^®^ 5X Plus Wearable Device at 4559 m

**DOI:** 10.3390/s21196363

**Published:** 2021-09-23

**Authors:** Lisa M. Schiefer, Gunnar Treff, Franziska Treff, Peter Schmidt, Larissa Schäfer, Josef Niebauer, Kai E. Swenson, Erik R. Swenson, Marc M. Berger, Mahdi Sareban

**Affiliations:** 1Department of Anesthesiology, Critical Care and Pain Medicine, Paracelsus Medical University, 5020 Salzburg, Austria; l.schiefer@hotmail.com (L.M.S.); f.treff@salk.at (F.T.); p.schmidt@salk.at (P.S.); l.schaefer@salk.at (L.S.); 2Ludwig Boltzmann Institute for Digital Health and Prevention, 5020 Salzburg, Austria; j.niebauer@salk.at; 3Division of Sports and Rehabilitation Medicine, University of Ulm, 89075 Ulm, Germany; gunnar.treff@uni-ulm.de; 4University Institute of Sports Medicine, Prevention and Rehabilitation and Research Institute of Molecular Sports Medicine and Rehabilitation, Paracelsus Medical University, 5020 Salzburg, Austria; 5Division of Pulmonary and Critical Care Medicine, Massachusetts General Hospital, Boston, MA 02114, USA; keswenson@mgh.harvard.edu; 6Division of Pulmonary, Critical Care, and Sleep Medicine, Beth Israel Deaconess Medical Center, Boston, MA 02215, USA; 7Division of Pulmonary, Critical Care and Sleep Medicine, VA Puget Sound Health Care System, University of Washington, Seattle, WA 98108, USA; erik.swenson@va.gov; 8Department of Anesthesiology and Intensive Care Medicine, University Hospital Essen, University Duisburg Essen, 45147 Essen, Germany; marc.berger@uk-essen.de

**Keywords:** hypoxia, altitude, accuracy

## Abstract

Decreased oxygen saturation (SO_2_) at high altitude is associated with potentially life-threatening diseases, e.g., high-altitude pulmonary edema. Wearable devices that allow continuous monitoring of peripheral oxygen saturation (SpO_2_), such as the Garmin Fēnix^®^ 5X Plus (GAR), might provide early detection to prevent hypoxia-induced diseases. We therefore aimed to validate GAR-derived SpO_2_ readings at 4559 m. SpO_2_ was measured with GAR and the medically certified Covidien Nellcor SpO_2_ monitor (COV) at six time points in 13 healthy lowlanders after a rapid ascent from 1130 m to 4559 m. Arterial blood gas (ABG) analysis served as the criterion measure and was conducted at four of the six time points with the Radiometer ABL 90 Flex. Validity was assessed by intraclass correlation coefficients (ICCs), mean absolute percentage error (MAPE), and Bland–Altman plots. Mean (±SD) SO_2_, including all time points at 4559 m, was 85.2 ± 6.2% with GAR, 81.0 ± 9.4% with COV, and 75.0 ± 9.5% with ABG. Validity of GAR was low, as indicated by the ICC (0.549), the MAPE (9.77%), the mean SO_2_ difference (7.0%), and the wide limits of agreement (−6.5; 20.5%) vs. ABG. Validity of COV was good, as indicated by the ICC (0.883), the MAPE (6.15%), and the mean SO_2_ difference (0.1%) vs. ABG. The GAR device demonstrated poor validity and cannot be recommended for monitoring SpO_2_ at high altitude.

## 1. Introduction

Wearable devices are increasingly used to monitor physiological biomarkers [1]. If certified by the relevant regulatory authorities, they can also be used to monitor and/or diagnose diseases, e.g., abnormal heart conditions, hypertension, and diabetes [2]. However, with the increasing demand for such devices, commercially available wearables may be used to assess the risk of potentially life-threatening diseases despite not being intended for such purposes and despite not having received medical certification. This means that these devices are being used without having undergone the rigorous validity testing required for certification as a medical device.

At high altitude, oxygen saturation (SO_2_) is reduced [3,4]. This condition is associated with a group of diseases categorized as high-altitude illnesses. These include mild diseases, like acute mountain sickness (AMS), but also potentially life-threatening diseases, such as high-altitude pulmonary edema (HAPE) [4]. Since peripheral oxygen saturation (SpO_2_) is a useful variable for evaluating an individual’s acclimatization status at high altitude and for monitoring the progression and treatment of high-altitude illnesses [3], valid and convenient SpO_2_ measurements are extremely desirable for mountaineers. It is worth noting that SpO_2_ is also useful for risk assessment and providing an early warning of deterioration in patients suffering from COVID-19 [5]. 

The criterion measurement for oxygen saturation (SO_2_) is arterial blood gas (ABG) analysis, which is an invasive and unpleasant procedure [6], requiring a needle puncture to a peripheral artery and subsequent analysis of a blood sample in a blood gas analyzer. This is generally not feasible during high-altitude sojourns, and therefore non-invasive medical devices are recommended for routine SpO_2_ measurements and risk assessment of high-altitude illness [7]. The transcutaneous fingertip-type pulse oximeter Covidien Nellcor SpO_2_ monitor (COV) is such a device. While these devices are simple to use, data output may be inaccurate [7]. They are also expensive, bulky, not suitable for continuous monitoring, and consequently not part of mountaineers’ regular equipment [8]. However, climbers commonly use wearables like smartwatches to track their performance and to guide and monitor their routes via GPS services [9]. Several recent commercially available smartwatches, including the Garmin Fēnix^®^ 5X Plus (GAR), include sensors for SpO_2_ measurement. Not surprisingly, they are increasingly being used by mountaineers with the intention of monitoring their risk of altitude illness [10], even though this usage has never been rigorously tested or confirmed.

While several studies have already investigated the validity and reliability of smartwatch-derived physiological biomarkers, such as heart rate and energy expenditure [11,12], data on the validity and reliability of SpO_2_ measurements are scarce [10,13,14]. Of note, there is a complete lack of data obtained in a high-altitude field setting. To this end, the aim of this study was to investigate whether GAR-derived SpO_2_ at 4559 m is valid by comparing it concurrently with COV and arterial oxygen saturation (SaO_2_) derived from the criterion measurement of an ABG sample. 

## 2. Methods

### 2.1. Study Approvals

The study was part of a prospective, randomized, placebo-controlled, double-blind study that investigated the effectiveness of acetazolamide in preventing high-altitude pulmonary edema. It was performed in accordance with the Declaration of Helsinki and its current amendments and was approved by the Ethics Committee of the Province of Salzburg, Austria; the Ethics Committee of the University of Turin, Italy; and by the Competent Authority (BASG), Vienna, Austria. Prior to inclusion in the study, all participants provided written informed consent.

### 2.2. Study Population

Thirteen native lowlanders were included in the study, which was carried out in 2019. All study participants met the predefined inclusion and exclusion criteria: all had a known history of high-altitude pulmonary edema (HAPE); none of the participants had spent time at altitudes >2000 m within four weeks before enrolling in the study; and none was found to have relevant medical illnesses during preliminary medical testing. Participants with concomitant cardiovascular diseases (other than well controlled systemic arterial hypertension) or pulmonary disease were excluded from the study.

### 2.3. Study Protocol

Baseline evaluations were performed at an altitude of 423 m (Salzburg, Austria). There, maximal cardiopulmonary exercise testing was conducted to assess aerobic capacity (⩒O_2max_). Participants completed a ramp test protocol on a cycle ergometer until voluntary exhaustion [15]. Depending on individual ability, the increment was chosen such that exhaustion occurred after 8–12 min. A breath-by-breath metabolic analyzer was used to measure gas exchange and ventilation (MetaLyzer 3B, Cortex Biophysics, Leipzig, Germany). 

Two to four weeks later, participants travelled to Alagna (1130  m), Valsesia, Italy, and ascended to 4559  m (Capanna Regina Margherita, Italy) within ~20 h, accompanied by licensed mountain guides. The ascent began with transport by cable car (from 1130 to 3275  m) and continued with a 90-min climb to the Capanna Giovanni Gnifetti (3611  m), where the participants spent the night. The next morning, they climbed to 4559  m (taking ~4 h), where they spent three nights and where all altitude measurements were conducted. The first examination took place between 17:00 and 19:00. The same examinations were repeated at 07:00 and 17:00 on each of the following two days; the last examination took place at 07:00 on the fourth day of the study. SpO_2_ was measured using GAR and the medically certified COV at six time points (at 6, 20, 30, 44, 54, and 68 h after arrival at 4559  m) and SaO_2_ was measured using ABG at four time points (at 20, 30, 44, and 68 h after arrival at 4559  m). ABG measurements were stopped in case of HAPE diagnosis and necessary medical treatment. 

### 2.4. Measurement of SO_2_

Garmin Fēnix^®^ 5X Plus (GAR)

GAR (Software version: 7.60.0.0) was used as outlined in the manufacturer’s instructions. The clean and dry watch was placed snugly but comfortably above the participant’s wrist bone at each measurement time point. The participants were asked to remain motionless in a supine position while the device read their SpO_2_ levels. After 5 min, the SpO_2_ value was noted.

Covidien Nellcor Portable SpO_2_ Patient Monitoring (COV)

Simultaneous to the measurement with the GAR, the reusable finger clip of the COV device (Nellcor PM10N, Covidien, Mansfield, USA) was applied on the other hand of the participant while they were supine and after 5 min the SpO_2_ value was noted.

Radiometer ABL 90 Flex

Arterial blood samples were collected from the participants after 10 min of rest using heparinized syringes equipped with a gold-coated mixing ball (safePICO, Radiometer, Brønshøj, Denmark). To ensure comparability between the three measurements, all were performed in the supine position, which is in any case obligatory for arterial blood sampling, and, moreover, as higher SpO_2_ values in the sitting position have already been reported [16]. Samples were immediately analyzed in triplicate using the blood gas analyzer Radiometer ABL 90 Flex (Radiometer, Brønshøj, Denmark) according to the manufacturer’s instructions. Triplicate measurements were averaged for further analysis, with the coefficient of variation of all triplicate measurements amounting to 0.82%. Measurements were conducted at the same time points as the noninvasive assessment with the GAR and COV devices.

Assessment of high-altitude illness

AMS was assessed at the time points of the COV and GAR measurements using the Lake Louise scoring (LLS) system and the self-administered, paper-based AMS cerebral (AMS-C) scoring system. AMS-C is an abbreviated version of the Environmental Symptoms Questionnaire III score [17,18,19]. AMS was diagnosed if LLS was ≥5 and an AMS-C score was ≥0.70. If only one of the two scores reached threshold values, the subject was classified as AMS-negative [20,21]. HAPE was diagnosed by daily chest radiography at high altitude.

## 3. Statistical Analysis

The following statistical procedures were applied to determine the validity of the results. A two-way mixed effects, absolute agreement, multiple raters/measurements intraclass correlation coefficient (ICC) [22] was calculated to assess validity, as suggested by De Vet et al. [23]. As suggested by Fokkema et al. [24], four thresholds were used to classify validity, as low (<0.60), moderate (0.60–0.75), good (0.75–0.90), or excellent (>0.90). In addition, a coefficient of correlation (Pearson’s r) and a coefficient of determination (R^2^) were calculated to examine associations between SO_2_ values derived from different methods and associations between SO_2_ values and AMS severity, as well as AMS and HAPE incidence. Correlational analysis of the differences between ABG and GAR results and the average between ABG and GAR results was used to test for magnitude dependence of any difference [25]. The mean absolute percentage error (MAPE) between measurements was calculated to provide a normalized measure of validity. The percentage accuracy of a model is calculated according to the equation: *MAPE = (1/n [sample size]) × Σ([actual data value]—[forecasted data value])/[actual data value]) × 100*. MAPE does not have a standardized threshold for determining the validity of measurements; however, Fokkema et al. considered a difference of ±5% as practically relevant for wearable sensor data [24]. Since repeated SpO_2_ testing in patients with chronic obstructive pulmonary disease showed intraday fluctuations of 1.6% [26], an SpO_2_ MAPE threshold ranging ±3.2% (i.e., two times the standard deviation (SD)) was assumed as a criterion of acceptable validity in our study. Furthermore, Bland–Altman analysis, including mean difference and limits of agreement, was conducted to plot the difference between the scores of two measurements against the mean SO_2_ values derived from the measurements [27]. Finally, data from the devices were compared using unpaired *t*-tests, with a *p*-value of 0.05 set as the threshold for significant differences.

Continuous data are given as arithmetic mean ± SD and categorical data as percentages. All statistical analyses were performed using SPSS 27 for Windows (SPSS, Inc., Chicago, IL, USA).

## 4. Results

Table 1 summarizes the baseline characteristics of the participants. 

Figure 1 displays the mean and SD values for SO_2_ measurements at all time points at high altitude. Overall, SO_2_ values obtained with GAR were highest (85.2% ± 6.2), compared to 81.0% ± 9.4 for those obtained with COV (*p* = 0.011) and 75.0% ± 9.5 for those obtained with ABG (*p* ≤ 0.001). Mean SO_2_ differences between GAR vs. ABG were 7.0%, compared to 0.1% between COV vs. ABG.

The ICC for GAR vs. ABG was low (0.549), whereas the ICC for COV vs. ABG was good (0.883). According to the predefined MAPE acceptable validity cut-off of <3.2%, neither GAR (9.77 %) nor COV (6.15 %) fulfilled this criterion when compared to ABG (Table 2). 

The Pearson’s r indicated a strong correlation between COV vs. ABG (0.904; *p* < 0.001; R^2^ = 0.735) and a weak correlation for GAR vs. ABG (0.380, *p* < 0.001; R^2^ = 0.109). The median of the individual Pearson’s r correlation coefficients between GAR and COV was 0.534, ranging from −0.5 to 0.909. Correlational analysis did not reveal a significant magnitude dependence of the difference between ABG and GAR and the average between ABG and GAR (r = 0.1, *p* = 0.625). Figure 2 shows the regression line of ABG vs. GAR in comparison to the line of identity. In three measurements with very low ABG results (SaO_2_ ≤ 70%), GAR did not report any SpO_2_ readings.

Bland–Altman analysis indicated a low validity of GAR measurements (mean difference compared to ABG: 7.0%) with wide limits of agreement (−6.5; 20.5%), whereas the validity of COV was good (mean difference compared to ABG: 0.1%) despite wide limits of agreement (−10.7; 10.9%) (Figure 3). 

The overall incidence of AMS was 77% (10/13) and the incidence of HAPE was 54% (7/13). The Pearson’s r showed a strong correlation between ABG and the AMS severity assessed by the Lake Louise Score (LLS) and a weak correlation between GAR and AMS severity (Table 3). 

## 5. Discussion

The aim of this study was to evaluate the validity of the Garmin Fēnix^®^ 5X Plus-derived SpO_2_ readings at 4559 m. The main result was the poor validity of GAR, indicated by an ICC of 0.549, an MAPE of 9.77, a mean SO_2_ difference of 7.0%, and wide limits of agreement (−6.5; 20.5%) vs. ABG. 

Wearable wrist-worn devices have the potential to serve as a convenient method to collect SpO_2_ data continuously and to improve health and safety during altitude activities through the early detection of lower SpO_2_ levels than expected at any given altitude. However, only a few validity studies of wearable SpO_2_ sensors exist. All of them were performed only at simulated altitude, using medical transcutaneous oximeters for criterion measurement with conflicting results. Lauterbach et al. evaluated the accuracy of SpO_2_ readings derived from the same Garmin device as used in the present study at simulated altitudes up to 3660 m. The authors concluded that GAR exhibits minimal overestimation (mean difference: 3.3%; limits of agreement: −1.9; 8.6%) of SpO_2_ and that the device may be a viable method to monitor SpO_2_ at high altitude [13]. However, in Lauterbach’s study, only Bland–Altman analysis was used to assess validity. More recently, Hermand et al. evaluated the accuracy of the Garmin Forerunner 245 SpO_2_ sensor in 10 healthy participants at simulated altitudes from 3000–5500 m [10] and applied more comprehensive statistical methods to assess device validity, including ICCs. The device failed to provide trustworthy SpO_2_ values, yielding an ICC of less than 0.280 over all the altitudes studied. Our study is the first to investigate the validity of GAR-derived SpO_2_ measurements in a field setting at 4559 m, comparing them with the standard criterion of arterial blood gas analysis, in addition to measurements obtained with a medically certified transcutaneous oximeter, and applying comprehensive statistical analyses. Compared to measurements taken at simulated altitudes in a hypoxic hypobaric chamber, SpO_2_ values at high altitude resemble real-life conditions, since they include the impact of environmental variables, such as cold and light, but also physiological variables, such as hyperventilation and periodic breathing, which may interfere with data stability [28,29]. Our multiparametric statistical analyses and predefined validity criteria demonstrate that GAR lacks acceptable validity, yielding a mean difference of 7.0% and an ICC of 0.549 compared to ABG. These results are similar to those of Hermand et al. (ICC < 0.280 over all simulated altitudes). 

There is increasing evidence that blood oxygenation is lower in individuals who suffer from AMS. Recently, an SpO_2_ threshold of 84% was reported for predicting the development of severe AMS with satisfactory specificity and sensitivity between 3600 to 3700 m using a 24-h data memory fingertip-type medical oximeter [30]. Based on our data, GAR overestimates SpO_2_ levels compared to ABG, with poor agreement indicated by wide limits of agreement in our Bland–Altman analysis (−6.5; 20.5%) and a low coefficient of determination (R^2^ = 0.109), which precludes using GAR to reliably categorize mountaineers with increased risk for AMS using the abovementioned cutoff value. Furthermore, in our study, GAR showed the lowest predictive value for assessing the severity of AMS (R^2^ = 0.007), whereas COV performed better (R^2^ = 0.278) and ABG yielded the best prediction (R^2^ = 0.644). 

Mountaineers suffering from HAPE often present with very low SpO_2_ levels due to impaired alveolar gas exchange at high altitude [31]. Although regression analysis did not reveal a significant magnitude dependence of the difference between ABG and GAR (*p* = 0.625), GAR tended to overestimate SpO_2_ especially when blood oxygenation was low. This is indicated by the relatively flat regression line of ABG vs. GAR in comparison to the line of identity (Figure 2). In addition, GAR was unable to measure any SpO_2_ values when ABG measurements were lowest. Added to the fact that taking measures to prevent altitude sickness is particularly important when O_2_ saturation is low, the overestimation of SpO_2_ by GAR at high altitude may falsely suggest that no risk is present. This does not only limit the usefulness of GAR but also means that reliance on its results could be potentially life-endangering. This notion is also in line with Luks and Swenson [32], who analyzed pulse oximetry for monitoring patients with COVID-19 at home in a recent focused review, as low SpO_2_ levels might also be an indicator for COVID-19-related pneumonia and adverse clinical outcome [33]. Easy to use and inexpensive, finger pulse oximeters can be considered an attractive option for monitoring COVID-19 patients at home; however, the authors raised awareness of the limited data on the accuracy of these devices, both for stand-alone finger oximeters and smart phone systems that do not have regulatory agency approval, especially when saturation falls below 90%. Additionally, in the future, the availability of new technologies, such as contact-less SpO_2_ analysis, e.g., via video processing, might add even more convenient methods to self-monitor SpO_2_ levels at high altitude [34], provided sound methodological studies prove their validity in real-world mountaineering conditions.

Our study has some limitations worth mentioning. SpO_2_ readings from wrist-worn devices might be influenced by skin tone, and this factor was not assessed in our study. However, there was no effect of skin tone on any SpO_2_ variables in the study of Hermand et al. [10]. Also, future firmware versions released by Garmin could change the accuracy of SpO_2_ measurements, and thus affect the conclusions of this study. We performed the measurements in a small sample of participants with a history of HAPE; a larger sample would have increased statistical power. Besides the logistical difficulties associated with high-altitude studies, the validity data obtained in this study should also be applicable to subjects without HAPE susceptibility, given the wide range of SpO_2_ reported here. 

In conclusion, SpO_2_ data obtained by the Garmin Fēnix^®^ 5X Plus at 4559 m do not meet acceptable validity criteria. Systematic overestimation of SpO_2_ levels at high altitude increases the probability that mountaineers misinterpret the risk of high-altitude illness, which has the potential to lead to life-threatening situations. Therefore, we cannot recommend GAR for monitoring SpO_2_ with the aim of acclimatization management or predictive health monitoring.

## Figures and Tables

**Figure 1 sensors-21-06363-f001:**
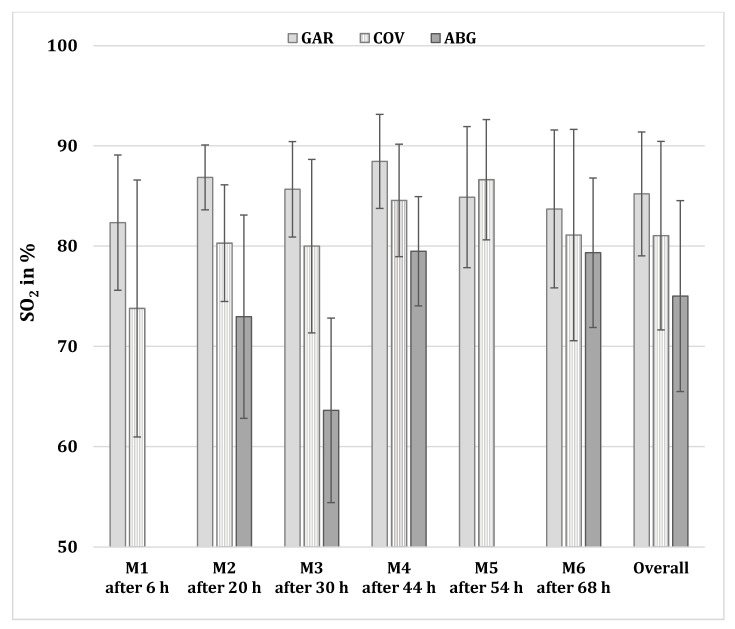
Mean SO_2_ values in percent (%) at high altitude (4559 m) at different time points after ascent. SO_2_ = peripheral/arterial oxygen saturation; GAR = Garmin Fēnix^®^ 5X Plus; COV = Covidien Nellcor Portable SpO_2_ Patient Monitoring; ABG = Radiometer ABL 90 Flex. Data given in mean ± SD.

**Figure 2 sensors-21-06363-f002:**
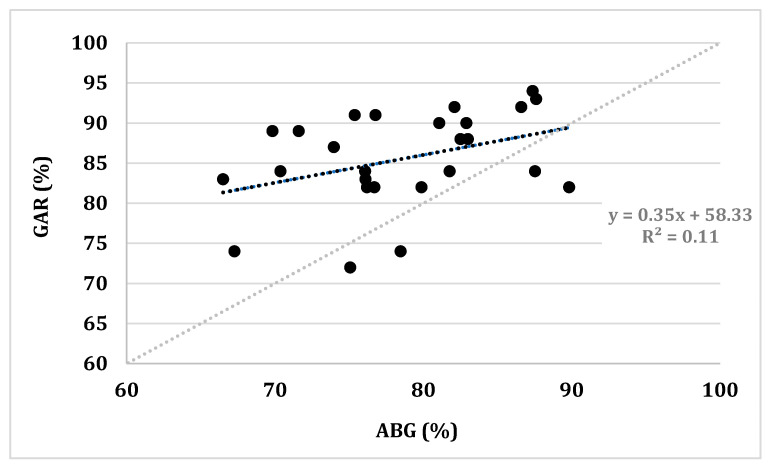
Scatterplot with regression line of ABG vs. GAR oxygen saturation in comparison to the line of identity. GAR = Garmin Fēnix^®^ 5X Plus; ABG = Radiometer ABL 90 Flex.

**Figure 3 sensors-21-06363-f003:**
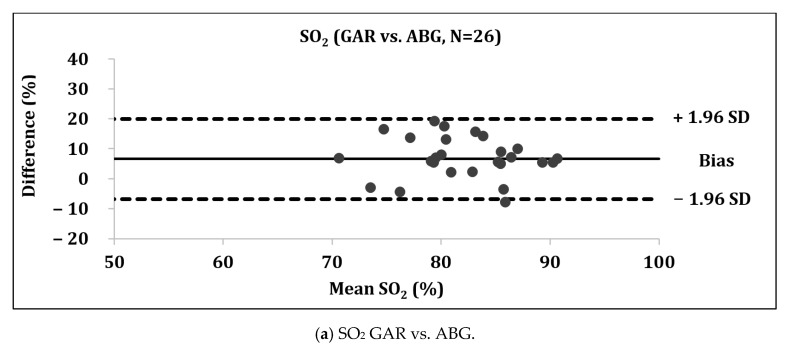
(**a**–**c**) Bland–Altman analysis with mean difference and limits of agreement. SO_2_ = peripheral/arterial oxygen saturation; GAR = Garmin Fēnix^®^ 5X Plus; ABG = Radiometer ABL 90 Flex. COV = Covidien Nellcor Portable SpO_2_ Patient Monitoring. Data given in mean ± SD.

**Table 1 sensors-21-06363-t001:** Anthropometric data of study participants (N = 13).

Sex	11 male, 2 female
Age (years)	57 ± 6
Body mass (kg)	76 ± 11
Body height (cm)	175 ± 7
Body mass index (kg/m^2^)	24.8 ± 3.3
⩒O_2max_ (ml/min/kg)	39 ± 9

⩒O_2max_ = maximal oxygen consumption. Data are presented as means ± SD.

**Table 2 sensors-21-06363-t002:** Validity criteria of GAR-derived SO_2_ values compared to those obtained with a medical device (COV) and with arterial blood gas analysis (ABG) at 4559 m.

	ICC	MAPE [%]	Pearson’s r	*p*-Value
GAR vs. COV (n = 49)	0.661	6.81	0.537	0.011 *
GAR vs. ABG (n = 37)	0.549	9.77	0.380	<0.001 *
COV vs. ABG (n = 26)	0.883	6.15	0.904	0.979

ICC = intraclass correlation coefficient; MAPE = mean absolute percentage error; GAR = Garmin Fēnix^®^ 5X Plus; COV = Covidien Nellcor Portable SpO_2_ Patient Monitoring; ABG = Radiometer ABL 90 Flex; * *p* < 0.05; *p*-values were derived via unpaired *t*-test.

**Table 3 sensors-21-06363-t003:** Correlation and linear regression analysis between SO_2_-derived variables from different devices and high-altitude illness.

Dependent Variable	SO_2_ Derived From	Pearson’s r	*p*-Value	R^2^
Severity LLS	GAR	−0.167	0.251	0.007
	COV	−0.541	<0.001 *	0.278
	ABG	−0.809	<0.001 *	0.644
AMS positive	GAR	0.073	0.618	−0.016
	COV	−0.123	0.399	−0.006
	ABG	−0.304	0.068	0.066
HAPE positive	GAR	−0.034	0.814	−0.020
	COV	−0.115	0.431	−0.008
	ABG	−0.345	0.036 *	0.094

SO_2_ = peripheral/arterial oxygen saturation; GAR = Garmin Fēnix^®^ 5X Plus; COV = Covidien Nellcor Portable SpO_2_ Patient Monitoring; ABG = Radiometer ABL 90 Flex; LLS = Lake Louise Score; AMS = acute mountain sickness; HAPE = high-altitude pulmonary edema; * *p* < 0.05.

## Data Availability

The data presented in this study are available on request from the corresponding author.

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
