# Peer review of "Validity of Peripheral Oxygen Saturation Measurements with the Garmin Fēnix® 5X Plus Wearable Device at 4559 m"

_sensors, 2021, doi:10.3390/s21196363_

Round 1

Reviewer 1 Report

The authors investigated whether the device GAR may accurately measure the SpO2 levels of persons on different altitudes. This is an interesting study and many readers may be of interest.

The cohort size is a little bit small. The authors mentioned in the Abstract that these 13 person are healthy persons. But actually these persons have known histories of HAPE. Another group of healthy persons without HAPE may be needed as the control samples, since the correlations of the GAR measurements and the gold standard are not very high. Maybe the healthy controls will have high correlations.

The correlations of all the samples are not very high. How about the gender-specific or person-specific correlations?

Author Response

We thank you for giving us the opportunity to revise our manuscript and for taking the time to review our manuscript. We appreciate the diligence in feedback, which helped us to improve its clarity and quality. Below, we respond to each comment in detail.

Replies to Reviewer# 1

The authors investigated whether the device GAR may accurately measure the SpO2 levels of persons on different altitudes. This is an interesting study and many readers may be of interest.

Comment 1. The cohort size is a little bit small. The authors mentioned in the Abstract that these 13 person are healthy persons. But actually these persons have known histories of HAPE. Another group of healthy persons without HAPE may be needed as the control samples, since the correlations of the GAR measurements and the gold standard are not very high. Maybe the healthy controls will have high correlations.

Response. Thank you for this comment. We use the term “healthy” to describe their health status during baseline examination and with reference to their medical eligibility to climb to 4,559 m. Furthermore, the aim of this study was not to compare SpO2-validity of GAR between mountaineers with a history of HAPE and mountaineers without. HAPE susceptible subjects are at increased risk for developing a life-threatening altitude-related sickness and present with a wide range of peripheral oxygen saturation. Thus, we consider this cohort, which for obvious reason is hard to recruit, as a strength of this study. As expected, following the rapid and active ascent to 4,559m, 54% of participants developed HAPE at high altitude. Notably, in table 3 we specifically present correlation and linear regression analysis between SO2-derived variables from different devices of participants who developed HAPE at high altitude. However, the reviewer is right and a larger sample size would have increased statistical power. We added the reviewers’ comment to the limitations of the study (line 303-307).

Comment 2. The correlations of all the samples are not very high. How about the gender-specific or person-specific correlations?

Response. Thank you for this valuable comment. We calculated person-specific correlations (Pearson’s r) of the different measurement time points, which yielded a median of 0.534, ranging from -0.5 to 0.909 for GAR vs. COV, which is similar to the correlation coefficient for the group-based analyses (0.537). We added this information to the manuscript (line 195-197). With regard to gender-specific correlations, only two female climbers participated in the study (Table 1). Thus, we decided against a gender-specific analyses. 

Of note, to improve clarification of the manuscript, we changed peripheral oxygen saturation (SpO2) to oxygen saturation (SO2) in case of referring to both, peripheral and arterial oxygen saturation.

Reviewer 2 Report

Authors evaluate the effectiveness of a commercial smartwatch for SPO2 monitoring in high altitude. In vivo experiments have been conducted. Statistical analyses has been used to compare the smartwatch capability of monitoring blood oxygen saturation with baseline instruments. 

The paper is well written and well organized. The experiments have been correctly conducted.

In the following some minor comments:

  • paragraph 3: please define and better describe the evaluation measures that have been used. This is needed to better understand the results
  • paragraph 2.4: please explain why measurements have been performed while patients were in a supine position
  • Figure 1: please reduce the dimension of the figure
  • Figure 3: please re-arrange the subfigures according with their description in the text.
  • In this article a contact-less device is used to asses the effectiveness of SPO2 measurements https://doi.org/10.1109/ISCC50000.2020.9219718

Author Response

We thank you for giving us the opportunity to revise our manuscript and for taking the time to review our manuscript. We appreciate the diligence in feedback, which helped us to improve its clarity and quality. Below, we respond to each comment in detail.

Replies to Reviewer# 2

Authors evaluate the effectiveness of a commercial smartwatch for SpO2 monitoring in high altitude. In vivo experiments have been conducted. Statistical analyses has been used to compare the smartwatch capability of monitoring blood oxygen saturation with baseline instruments. This paper is well written and well organized. The experiments have been correctly conducted.

Comment 1. Paragraph 3: please define and better describe the evaluation measures that have been used. This is needed to better understand the results.

Response. We thank the reviewer for pointing out this issue. We extended the description of the statistical analyses section, as suggested (line 155-157 and 164-165).

Comment 2. Paragraph 2.4: please explain why measurements have been performed while patients were in a supine position

Response. Thank you for this comment. Since the criterion measurement of this study, i.e. arterial blood drawing, needs to be performed in a supine position, all measurements were performed in the same position to ensure comparability between devices. We added this information to the methods section of the manuscript (line 128-130).

Comment 3. Figure 1: please reduce the dimension of the figure.

Response. We have reduced the dimension of the figure as suggested.

Comment 4. Figure 3: please re-arrange the subfigures according with their description in the text.

Response. We have re-arranged the subfigures as suggested.

Comment 5. In this article a contact-less device is used to assess the effectiveness of SPO2 measurements (https://doi.org/10.1109/ISCC50000.2020.9219718).

Response. Thank you for pointing out this interesting article. We have added this novel approach and the reference in the manuscript (line 295-298), which might be of interest in future for mountaineers who aim to assess their peripheral oxygen saturation in an indoor setting with their smart phone. However, considering that environmental conditions at high altitude such as cold, light and wind may impact this approach, methodological studies need to prove its validity at real-world mountaineering conditions.

Of note, to improve clarification of the manuscript, we changed peripheral oxygen saturation (SpO2) to oxygen saturation (SO2) in case of referring to both, peripheral and arterial oxygen saturation.